# *O*-methylated N-glycans Distinguish Mosses from Vascular Plants

**DOI:** 10.3390/biom12010136

**Published:** 2022-01-15

**Authors:** David Stenitzer, Réka Mócsai, Harald Zechmeister, Ralf Reski, Eva L. Decker, Friedrich Altmann

**Affiliations:** 1Department of Chemistry, Institute of Biochemistry, University of Natural Resources and Life Sciences, Vienna, Muthgasse 18, 1190 Vienna, Austria; david.stenitzer@boku.ac.at (D.S.); reka.mocsai@boku.ac.at (R.M.); 2Department of Botany and Biodiversity Research, University of Vienna, Rennweg 14, 1030 Vienna, Austria; harald.zechmeister@univie.ac.at; 3Plant Biotechnology, Faculty of Biology, University of Freiburg, Schaenzlestrasse 1, 79104 Freiburg, Germany; ralf.reski@biologie.uni-freiburg.de (R.R.); eva.decker@biologie.uni-freiburg.de (E.L.D.)

**Keywords:** moss, bryophytes, glycoprotein, N-glycan, methyl-mannose

## Abstract

In the animal kingdom, a stunning variety of N-glycan structures have emerged with phylogenetic specificities of various kinds. In the plant kingdom, however, N-glycosylation appears to be strictly conservative and uniform. From mosses to all kinds of gymno- and angiosperms, land plants mainly express structures with the common pentasaccharide core substituted with xylose, core α1,3-fucose, maybe terminal GlcNAc residues and Lewis A determinants. In contrast, green algae biosynthesise unique and unusual N-glycan structures with uncommon monosaccharides, a plethora of different structures and various kinds of *O*-methylation. Mosses, a group of plants that are separated by at least 400 million years of evolution from vascular plants, have hitherto been seen as harbouring an N-glycosylation machinery identical to that of vascular plants. To challenge this view, we analysed the N-glycomes of several moss species using MALDI-TOF/TOF, PGC-MS/MS and GC-MS. While all species contained the plant-typical heptasaccharide with no, one or two terminal GlcNAc residues (MMXF, MGnXF and GnGnXF, respectively), many species exhibited MS signals with 14.02 Da increments as characteristic for *O*-methylation. Throughout all analysed moss N-glycans, the level of methylation differed strongly even within the same family. In some species, methylated glycans dominated, while others had no methylation at all. GC-MS revealed the main glycan from *Funaria hygrometrica* to contain 2,6-*O*-methylated terminal mannose. Some mosses additionally presented very large, likewise methylated complex-type N-glycans. This first finding of the methylation of N-glycans in land plants mirrors the presumable phylogenetic relation of mosses to green algae, where the *O*-methylation of mannose and many other monosaccharides is a common trait.

## 1. Introduction

The many branches of the tree of life distinguish themselves—inter alia—by their characteristic sets of structural features of N-glycans. Chordates have evolved sialic acids, which are missing in all other parts of the eukaryotic world as far as we currently know [1,2]. Subtle changes in the general N-glycome of vertebrates have occurred in the course of the development of vertebrates; thus, the structural repertoires of fish, birds, mammals and humans differ. Protostomia such as worms, molluscs, or arthropods impress with a variety of yet other structural peculiarities, such as phosphoethanolamine or substituted fucose residues, just to name a few examples [1]. Plants, however, appear to be highly conservative, displaying the very same set of N-glycan structures throughout the division of land plants, as was revealed by survey studies on the N-glycomes of pollen and food allergens [3,4]. The plant N-glycome was deciphered in two waves. The first one culminated in the late 1970s with the elucidation of the exact structure of the N-glycan of the pineapple stem protease bromelain [5,6]. This pioneering work was soon followed by the elucidation of the structure of the horseradish peroxidase N-glycan [7] and examples of terminal *N*-acetylglucosamine residues [3,8,9,10]. Here, it could be included that the xylose and even more the core α1,3-fucose residue are recognised as immunogenic and even allergy-relevant determinants [7,11,12,13].

A second wave of discovery revealed decoration with Lewis A determinants on one or two antennae [14,15]. This actually widely distributed feature [3,8,10,14] was overlooked for some time as it can hardly ever be found on high-abundance storage proteins but rather on secretory or cell wall glycoproteins [14]. The ultimate conservation of these traits points to an essential physiological function of complex-type plant glycans, which, however, have not been fathomed so far as the deletion of either xylosyl-transferase, core-α1,3-fucosyltransferase, or the Lewis A fucosyltransferase did not result in obvious phenotypic alterations [16,17,18,19]. Even a GlcNAc-transferase I-deficient *Arabidopsis* line, at first glance, appeared as a normal viable plant [20,21], but for tomatoes, complex glycans do play a vital role [22].

The structural investigation of the biotechnologically relevant moss species *Physcomitrella patens* (now *Physcomitrium patens*) [23] by HPLC or (low-resolution) MALDI-TOF MS displayed a picture exactly matching that seen with apple, onion, cauliflower, or any other fresh plant tissue [10]. It was the incidental appearance of an impressive amount of sporangia on a lawn of moss outside the author’s living room that enticed us to record yet another boring plant N-glycome. However, this moss—*Bryum caespiticium*—looked different. Now viewed with high-resolution MALDI-TOF MS, the small complex-type glycans MMXF and MGnXF (or, more precisely, MMXF^3^ and MGnXF^3^) were accompanied by prominent peaks each larger by 14.02 mass units, strongly indicative of methylation (Figure 1).

In this work, we elaborate on this observation by investigating the N-glycomes of several other bryophytes and determine the exact structure of two examples of methylated complex-type N-glycans from moss. Some mosses additionally exhibited N-glycans of yet inexplicable composition. The particular moss lawn, however, that initiated this research disappeared and was never seen again.

## 2. Materials and Methods

### 2.1. Materials

Biological samples were received either as axenic cell cultures from the International Moss Stock Centre (IMSC, www.moss-stock-center.org, accessed on 11 January 2022), collected in nature (*Calliergonella cuspidata*, *Bryum caespiticium*, *Plagiomnium undulatum*, *Polytrichum formosum*, *Sphagnum capillifolium* and *Rhytidiadelphus squarrosus*), from cell culture (*Aloina aloides* (#40087), *Brachythecium rutabulum* (#41268), *Ceratodon purpureus* (#40085), *Funarica hygrometrica* (#40017, #40086), *Grimmia pulvinata* (#41109), *Hypnum cupressiforme* (#40073), *Physcomitrium patens* (formerly *Physcomitrella patens*, #41269), *Physcomitrium eurystomum* (#40052), *Physcomitrium sphaericum* (#40043) and *Tortula muralis* (#41245)), or from the Obi supermarket aquarium division (*Amblystegium serpens* and *Vesicularia montagnei*). Mosses deriving from field sampling were identified by H. G. Zechmeister following the systematic classification proposed by Hodgetts et al. [23].

### 2.2. N-glycan Isolation and Fractionation

Fresh moss samples were treated with pepsin followed by the capture of peptides and glycopeptides by a cation exchange resin. After glycopeptide enrichment by size-exclusion chromatography, N-glycans were released by PNGase A and recovered by another cation exchange step as detailed previously [24,25].

The purified N-glycans were used for MALDI- and ESI-MS/MS [24,25]. The reduction of the glycans was carried out in 50 mM of NaOH with 1% NaBH_4_. The reduced samples were purified using a PGC-SPE column (10 mg).

For GC-MS measurements the glycans were fractionated by HILIC equipped with a TSK Amide-80 column (4 × 250 mm, 5 µm; Tosoh Bioscience GmbH, Griesheim, Germany). Fractions of 1 mL were analysed by MALDI-TOF-MS. Fractions containing mainly the glycan of interest were used for GC-MS analysis [24,25].

### 2.3. Mass Spectrometric Methods

For MALDI-TOF MS analyses, 1 µL of the sample was dried on the MALDI target plate, followed by 1 µL of 2% solution of 2,5-dihydroxybenzoic acid in 50% acetonitrile, which was used as matrix. Spectra were obtained with a Bruker Autoflex MALDI-TOF instrument in the positive ion reflectron mode. In special cases, glycans were permethylated prior to MALDI-TOF MS using dimethyl sulfoxide, solid NaOH and methyl iodide [26].

For linkage analysis by gas chromatography-mass spectrometry (GC-MS), oligosaccharides were permethylated with iodomethane-d3. Subsequently, they were hydrolysed at 100 °C with 2 M TFA and reduced with NaBD_4_. After peracetylation, the monosaccharides were loaded onto a GC-MS system (GC 7820 A & MSD 5975, Agilent, Waldbronn, Germany) equipped with an Agilent J&W HP-5ms GC column (30 m × 0.25 mm, 0.25 µm). Retention times of partially methylated alditol acetates were established by standards. [24,25].

For the PGC-LC-ESI-MS/MS analysis, the samples were separated with an LC equipped with a capillary column (Hypercarb, 100 mm × 0.32 mm, 5 μm particle size; Thermo Scientific, Waltham, MA, USA) coupled to an ion trap instrument (amaZon speed ETD; Bruker, Bremen, Germany). The starting conditions were 99% solvent A (10 mM ammonium bicarbonate) and 1% solvent B (80% acetonitrile in solvent A). From 4.5 min to 5.5 min, the concentration of solvent B was increased to 9%. From minutes 20 to 38, solvent B was increased to 24% and then to 30%, until 41.5 min.

MS scans were recorded from 600 to 1500 *m/z* with an ion trap (amaZon speed ETD; Bruker, Bremen, Germany) in negative mode, with the ICC target set to 100,000, the maximum accumulation time set to 250 ms and the target mass to 650 *m/z*.

For MS/MS measurements, 3 precursor ions with active exclusion were used. Fragmentation time was set to 35 ms and FxD was enabled.

### 2.4. Phylogenetic Analysis

The phylogenetic tree was generated using the online tool PhyloT (https://phylot.biobyte.de/, accessed on 3 December 2021). Since the species *Aloina aloides* was not contained in the database, the entry for the genus *Aloina kindb.* was used instead.

## 3. Results

### 3.1. Discovery of Methylation of Moss N-glycans

The MALDI-TOF MS spectrum of the released N-glycans from sporangia as well as gametophytes of *Bryum caespiticium* exhibited eye-catching extra peaks next to the usual complex-type plant N-glycans (Figure 1A). The glycosylation pattern of sporangia and lawn did not show a significant difference (data not shown).

The exact mass increment of 14.02 pointed at a methylation of the glycans, rather than another modification (Appendix A). After the first sign that methylated N-glycans may also exist in land plants, the question appeared, if this moss was one of its kind or if there are methylated N-glycans in other moss species as well.

### 3.2. N-glycosylation Pattern of Different Moss Species

After purifying the N-glycans from several moss species, the glycan patterns were analysed with MALDI-TOF (Appendix A). The main masses found represented the glycans MMXF, MGnXF and GnGnXF as well as singly and doubly methylated versions of these (Figure 1 and Figure 2). The main glycan type in many analysed moss species represented the mass of MMXF, while two of the analysed species had a main glycan that was doubly methylated (Figure 2). The moss *Funaria hygrometrica* presented a doubly methylated MGnXF (1442.53 Da [M+Na]^+^) as the main glycan (Figure 1B and Appendix A), whereas doubly methylated MMXF (1239.43 Da [M+Na]^+^) dominated in *Plagiomnium undulatum* (Figure 1C and Appendix A). In many mosses with methylated N-glycans, singly methylated MMXF and MGnXF were more abundant than the doubly methylated glycans. However, the mosses with higher methylation levels, such as *Funaria hygrometrica*, *Tortula muralis*, *Ceratodon purpureus*, *Plagiomnium undulatum* and *Bryum caespiticium*, tended to have a higher abundance of doubly methylated glycans compared to singly methylated ones. *Hypnum cupressiforme*, *Vesicularia montagnei* and *Calliergonella cuspidata*, all belonging to the order of Hypnales, had a higher abundance of singly as compared to doubly methylated glycans. For the glycan GnGnXF, only singly methylated glycans were found in all moss species with methylation. Moreover, GnGnXF consistently carried the lowest amount of methylation. The degree of methylation varied immensely between different moss species, even in mosses belonging to the same family. For this, examples are the mosses *Aloina aloides* and *Tortula muralis*, both belonging to the family of *Pottiaceae*. While *Tortula muralis* had a high amount of methylated glycans, *Aloina aloides* showed no methylation at all (Appendix A). Moreover, in the family of *Funariaceae*, the methylation level was heterogeneous—e.g., MGnXF was highly methylated only in *Funaria hygrometrica*. *Physcomitrium sphaericum* and Physcomitrium eurystomum also had methylated N-glycans, but the amount was low (Figure 2). Of all moss species analysed, the third *Physcomitrium* species, *P. patens*, was striking for its predominant (unmethylated) complex-type GnGnXF glycosylation. How far the methylation patterns are stable during the life cycle of the mosses or reflect different stages is currently unknown.

To obtain more insight into the relations between the analysed mosses, a phylogenetic tree was built with an online tool. The resulting tree nicely matched a recently published taxonomy [27]. Although the phylogenetic tree showed clustering, these clusters did not match the methylation status. The only cluster that showed homogeneous methylation was the cluster with *Plagiomnium undulatum* and *Bryum caespiticium*, both belonging to the order of *Bryales*. The other clusters were very heterogeneous in their level of methylation, which is comparable to Figure 2 (Figure 3). The samples derived from mosses adapted to both arid and highly humid habitats. They comprised gametophytes and, in case of *Bryum*, also sporophyte tissue. Insights into the regulation and possible biological function of N-glycan methylation have to await more systematic, ideally prospective studies.

### 3.3. Structural Analysis of the Main Glycans from Funaria hygrometrica and Plagiomnium undulatum

For structural analysis, the glycomes of *F. hygrometrica* and *P. undulatum* were fractionated with HILIC-HPLC to purify the doubly methylated MGnXF from *F. hygrometrica* and the doubly methylated MMXF from *P. undulatum*. The fractions containing mainly the glycans of interest were then hydrolysed, reduced and peracetylated before loading them on a GC-MS. For *F. hygrometrica* one part of the sample was also permethylated with iodomethane-d3, to get insight into the linkages of the monosaccharides and to confirm the overall structure of what was assumed to be the typical plant glycan MGnXF. The measurement showed that the glycan indeed adopted the well-known plant-type MGnXF structure with a doubly methylated, terminal mannose (Appendix A). The MS fragments located one methyl group to the C2, but gave an unclear result for the second one. A clearer identification was obtained for both the major N-glycans of *F. hygrometrica* and *P. undulatum* by the omission of the permethylation step. Fragmentation pattern (Appendix A) and retention time (Appendix A) clearly identified a 2,6-di-*O*-methyl mannose. To contest the assumption that this methyl mannose represented the 6-branch, 2me-MMXF and 2me-MGnXF were subjected to negative mode collision-induced decay in an LC-ESI-MS analysis. D-ions of *m/z* 351 in both glycans established the 6-arm as carrying the methyl-mannose [28].

The structure of the *F. hygrometrica* glycan is therefore MGnXF with a doubly methylated mannose on the upper arm. The structure of the main glycan from *P. undulatum* is an MMXF with a doubly methylated mannose on the upper arm (Figure 4). The arm location of methyl mannose was substantiated by the D-ion observed in negative mode collision-induced dissociation (Figure 5).

### 3.4. Other Glycan Masses

Finally, an additional feature of some of the mosses shall not go unmentioned. Peaks in the mass range 1800 to 2600 Da could only in part be explained as complex-type plant N-glycans with Lewis A determinants (*m/z* 1925 and 2233) (Figure 6). Peaks of *m/z* 1882, 2085 and 2553 apparently had displaced the familiar *m/z* 1925 and 2233 glycans in *Amblystegium serpens*, *Ceratodon purpureus*, *Hypnum cupressiforme*, *Plagiomnium undulatum*, *Rhytidiadelphus squarrosus* and *Vesicularia montagnei* (Appendix A). The mass shift induced by reduction with NaBH_4_ verified the glycan nature of these peaks. Permethylation of the *m/z* 2553 peak resulted in a mass of 3100.8 *m/z* (Appendix A), which translates into 5 hexoses, 4 HexNAcs, 5 fucoses and 1 pentose. The large moss glycan was, therefore, tentatively interpreted as resulting from the known Lewis A glycans by the addition of fucose and methyl residues (Figure 6).

## 4. Discussion

The phylogenetic position of mosses is often seen as intermediate between vascular plants and algae; in fact, the genome of the model moss *P. patens* encodes genes that have homologs in algae or fungi, but not in vascular plants and vice versa [29,30]. Here, we found a similar situation regarding their protein N-glycosylation. The glycan structures from vascular plants can mostly be found in mosses, while the methylation on these glycans is still something that remains from the algal past. The extreme diversity of glycan structures that is found in algae is, however, not discovered in mosses. Except for methylation and the occurrence of extra-large glycans, the moss N-glycan patterns are very similar and are based on the same structures also found in vascular plants.

The main glycans from *Funaria hygrometrica* and *Plagiomnium undulatum* both show terminal 2,6-*O*-methylated mannose. In the case of *F. hygrometrica*, the glycan structure is an MGnXF, while the one from *P. undulatum* is MMXF with 2 methyl groups. The competition for the 6-arm mannose’s 2-position, by either a methyl group or by GlcNAc, explains why there is no doubly methylated GnGnXF in any moss. GnGnXF is methylated to a very small degree and, if at all, carries only one methyl group (Figure 2). It is tempting to associate methylation of mannose in moss with that in the alga *Chlorella vulgaris*, but the differences are huge. First, *C. vulgaris* puts methyl groups to the 3 or 6 positions rather than 2 or 6; second, *C. vulgaris* methylates oligomannosidic glycans, while mosses apparently require the prior action of GlcNAc-transferase I, as methylation occurs on the mannose residues that become accessible only after the action of GlcNAc-transferase I and Golgi α-mannosidase II in the canonical processing pathway.

Different species displayed very different degrees of mannose methylation and we shall concede that the data collected in this study must be seen as snapshots taken at an essentially arbitrary physiological or replicatory state of the respective species. Much will have to be learned about the biosynthesis and responsible enzymes; the immunogenicity; and, last but not least, the physiological role of mannose methylation.

The methylation of small N-glycans is a structural feature found in algae [31] and molluscs [32] but hitherto unknown to occur in land plants. On top of that, some mosses contain very large N-glycans of, as yet, undefined structures, which possibly are extensions of the Lewis A-containing complex-type structures.

## 5. Conclusions

Bryophyta, widely known as mosses, essentially exhibit the same N-glycosylation machinery as vascular plants. However, in the form of methylation of the 6-arm mannose and the occurrence of unusual elongations of large complex-type glycans, several mosses distinguish themselves by hitherto undetected features.

## Figures and Tables

**Figure 1 biomolecules-12-00136-f001:**
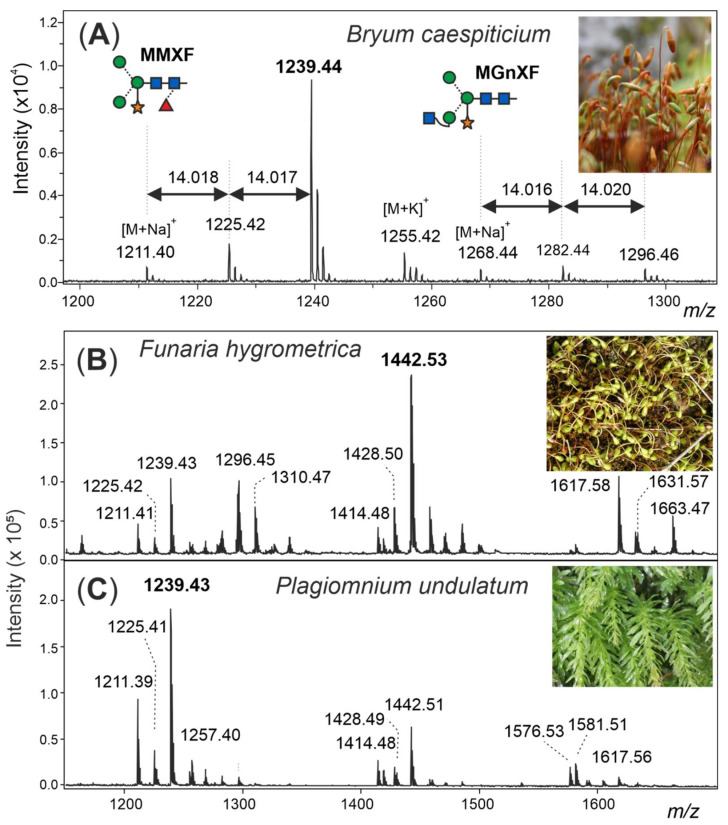
Demonstration of methylation of moss N-glycans. The structure abbreviations MMXF and MGnXF specify the terminal residues in a counter-clockwise manner (see also Appendix A). (**A**) Detail of the MALDI-TOF MS spectrum of N-glycans from the sporangia of *Bryum caespiticium* showing 14.02 Da increments. Section of the MALDI-TOF spectra of N-glycans from *Funaria hygrometrica* (**B**) and *Plagiomnium undulatum* (**C**) showing the different dominant complex-type N-glycans and their methylation. Complete spectra of all mosses analysed can be found in the Supporting Information (Appendix A).

**Figure 2 biomolecules-12-00136-f002:**
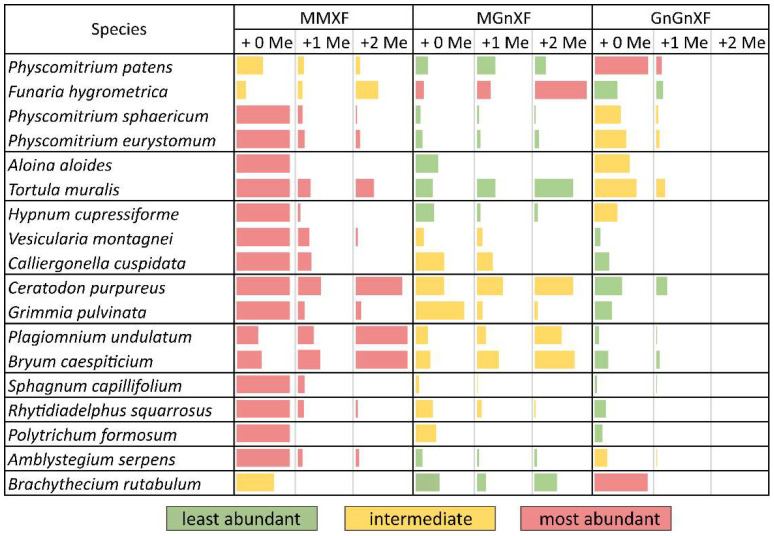
Glycan abundance in different moss species. Mosses from the same family are clustered together in separate boundaries. The abundance of MMXF, MGnXF and GnGnXF with their methylation is shown with coloured bars. The size of the bars is relative to the most abundant glycan from each species. Red indicates the most abundant glycan form. *Physcomitrium patens* was formerly known as *Physcomitrella patens*.

**Figure 3 biomolecules-12-00136-f003:**
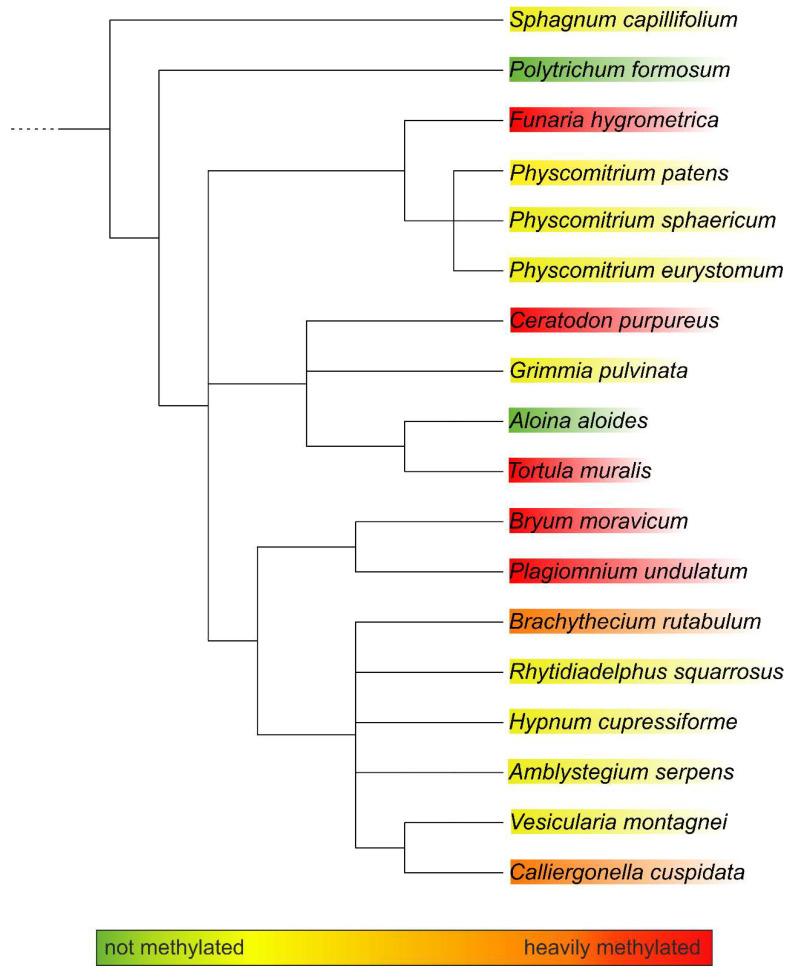
Phylogenetic tree of the analysed moss species as generated with PhyloT. The degree of methylation is indicated by the colour code.

**Figure 4 biomolecules-12-00136-f004:**
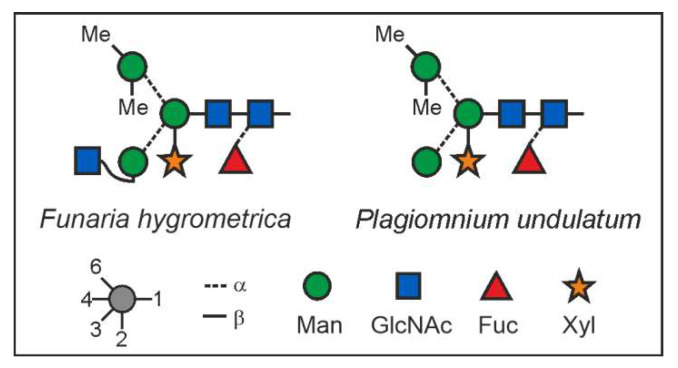
N-glycan structures of the main glycans from *Funaria hygrometrica* and *Plagiomnium undulatum.*

**Figure 5 biomolecules-12-00136-f005:**
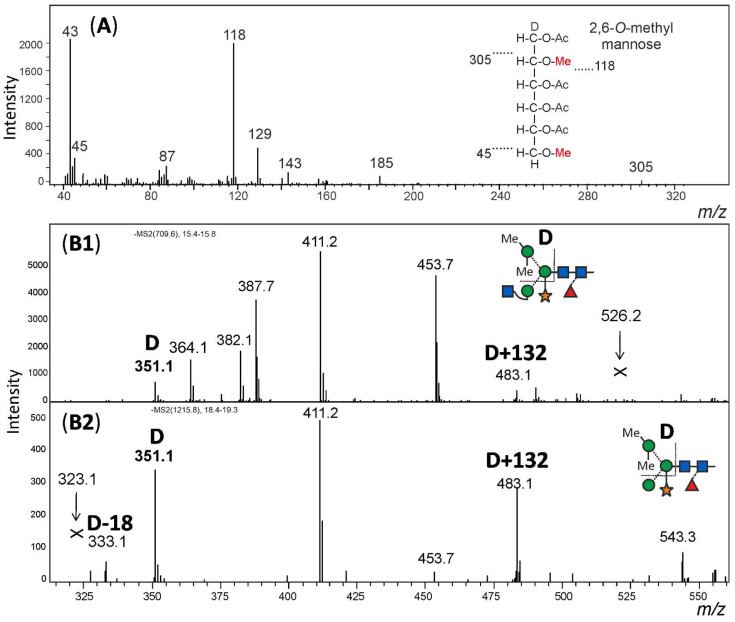
Structural analysis of methylated N-glycans. Panel (**A**): The HILIC fractionated main glycan from *Funaria hygrometrica* was hydrolysed, reduced and peracetylated. The fragment peaks in GC-MS analysis identified the methyl mannose as 2,6-di-*O*-methylation, which fits the retention time of that peak (Appendix A). Panels (**B1**,**B2**): The arm location of the methylated mannose was defined by observing the D-ion generated by negative mode collision induced dissociation. Notably, the xylose substituting the β-mannose residue was partially retained on the D-ions. Arrows pointing at Andrew’s crosses denote the theoretical positions of D-ion for the alternative branch arrangement.

**Figure 6 biomolecules-12-00136-f006:**
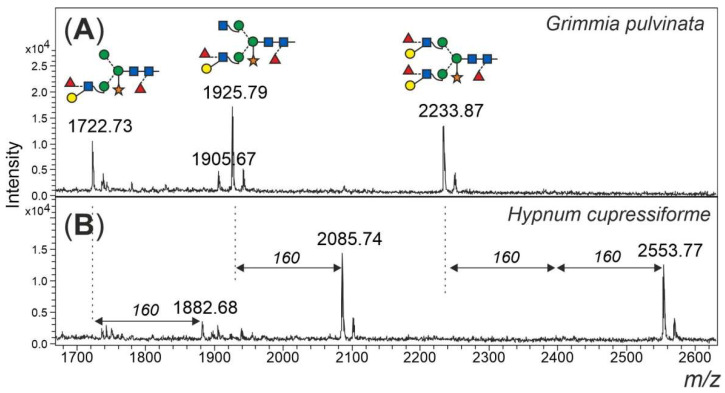
Large, unusual N-glycans occurring in some mosses. Panel (**A**) gives the high mass region of the moss *Grimmia pulvinata* with regular complex-type plant N-glycans. Panel (**B**) depicts the high mass region of the N-glycan pattern observed in a few moss species such as *Hypnum curessiforme*.

## Data Availability

The data presented in this study are—as far as not contained in the Appendix A—available upon request from F.A.

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
