# Peer review of "O-methylated N-glycans Distinguish Mosses from Vascular Plants"

_biomolecules, 2022, doi:10.3390/biom12010136_

Round 1

Reviewer 1 Report

The paper reads well, has clear results and conclusions.

I only have a few spelling things and an issue in figure 4.

Pager 3 line 100: are = were

Page 3 line 113: a = an

Page 3 line136: differnece =difference

Page 5 Figure 2 legend: clustered together in a separate w.  What is w? window?

Page 7 figure 4: in the text you speak of 2me-MMXF and 2me-MGnXF, but in figure 4B1 and B2, there is no fucose on the glycan. Although not that important for the fragment spectrum, I think you should add a fucose to the glycan cartoon. I think the indicated masses correspond to structures with fucose.

Author Response

Reply to reviews

A: We thank the reviewer for her/his serious and thorough reading of the manuscript and the helpful comments.

Reviewer I:

Comments and Suggestions for Authors

The paper reads well, has clear results and conclusions.

I only have a few spelling things and an issue in figure 4.

Pager 3 line 100: are = were

A: Please excuse, but here we cannot fully agree

Page 3 line 113: a = an

A: we cannot fully agree.

Page 3 line136: differnece =difference

A: corrected

Page 5 Figure 2 legend: clustered together in a separate w.  What is w? window?

A: We thank the reviewer for pointing out this mistake.

Page 7 figure 4: in the text you speak of 2me-MMXF and 2me-MGnXF, but in figure 4B1 and B2, there is no fucose on the glycan. Although not that important for the fragment spectrum, I think you should add a fucose to the glycan cartoon. I think the indicated masses correspond to structures with fucose.

A: We thank the reviewer for detecting this embarrassing error.

Reviewer 2 Report

The manuscript is interesting and covers an under explored field, such as moss glycosylation, leading to an inspiring conclusion, which may awake the scientific interest in this group of plants resulting in novel research projects and findings.

The introduction is concise but covers the required essential information to understand the manuscript.

The materials and methods are conventional and well described but there is information missing regarding the taxonomic classification of the mosses collected from nature, starting from the moss that originated the study. The authors should give details on who/how/where did the identification and classification of the diverse species.

The results section requires a thorough revision and some rewriting, though no additional experiments are needed. Influence of the origin of each sample should be taken into consideration and if found relevant, discussed, as there are samples from nature, cell cultures and axenic cell cultures, which may influence the glycosylation patterns observed. Some of the conclusions and statements in point 3.2 (N-glycosylation pattern of different moss species) are not really accurate and should be described appropriately (i.e. “most” is not 5 out of 9 mosses; the “higher abundance for singly methylated glycans” is not that obvious. It is unclear the statement in lines 174-176 (closely related to what?).

The glycan nomenclature used is only explained in Figure S1, which is not even cited in the main text. Since this is not a widely used nomenclature for glycans, references should be included and the nomenclature should be briefly explained within the main text. However, since this nomenclature omits important information on the glycan structures, I suggest to include a table with the detailed glycan sequences in a conventional format stating all the linkages and positions. This is important for a non-expert on plant glycosylation public.

Figure legends need to be revised and extended, especially those of figures 2 and S1 and Table S1. Figures, tables and figure panels should be accurately cited (i.e. fig 1A in line 135 instead of Figure 1; Table?//Figure2 in line 159; figure 1 panels B and C in lines 160 and 161 respectively), figure S2 A and B are never cited.  

It would be good to briefly discuss the distribution of the selected mosses within the phylogenetic tree, for the reader to understand how representative are the results obtained. For example, how representative is the finding that the closely related Physcomitrella and Physcomitrium or Bryum and Plagiomnium exhibit similar methylation? Stating that the Pottiaceae family is an “extreme example” is not far too much considering that only two species of mosses of that family have been included in the study or they are really representative of the family?

References for the information regarding Chlorella vulgaris should be included (lines 280-283 of the discussion).

Relevance for the possible Lewis A containing complex type glycan sequences should be discussed.

Author Response

Reply to reviews

A: We thank the reviewer for her/his serious and thorough reading of the manuscript and the helpful comments.

Reviewer II:

Comments and Suggestions for Authors

The manuscript is interesting and covers an under explored field, such as moss glycosylation, leading to an inspiring conclusion, which may awake the scientific interest in this group of plants resulting in novel research projects and findings.

The introduction is concise but covers the required essential information to understand the manuscript.

The materials and methods are conventional and well described but there is information missing regarding the taxonomic classification of the mosses collected from nature, starting from the moss that originated the study. The authors should give details on who/how/where did the identification and classification of the diverse species.

A: A line has been added in the Materials and Methods section stating who did the classification based on which work.

The results section requires a thorough revision and some rewriting, though no additional experiments are needed. Influence of the origin of each sample should be taken into consideration and if found relevant, discussed, as there are samples from nature, cell cultures and axenic cell cultures, which may influence the glycosylation patterns observed. Some of the conclusions and statements in point 3.2 (N-glycosylation pattern of different moss species) are not really accurate and should be described appropriately (i.e. “most” is not 5 out of 9 mosses; the “higher abundance for singly methylated glycans” is not that obvious. It is unclear the statement in lines 174-176 (closely related to what?).

A: These are several helpful hints: First, “most” was changed to “many”. Second: “had a higher abundance of singly as compared to doubly methylated glycans”should be clearer now, and Third, the “closely related” has been re-formatted.

The glycan nomenclature used is only explained in Figure S1, which is not even cited in the main text. Since this is not a widely used nomenclature for glycans, references should be included and the nomenclature should be briefly explained within the main text. However, since this nomenclature omits important information on the glycan structures, I suggest to include a table with the detailed glycan sequences in a conventional format stating all the linkages and positions. This is important for a non-expert on plant glycosylation public.

A: This is definitely a good point. We added the sentence “The structure abbreviations MMXF and MGnXF specify the terminal residues in a counter-clockwise manner (see also Figure S1).” to the Legend of Figure 1 and a hint to Fig. 1 in the main text. This should clarify the matter without excessively expanding the text.

Figure legends need to be revised and extended, especially those of figures 2 and S1 and Table S1. Figures, tables and figure panels should be accurately cited (i.e. fig 1A in line 135 instead of Figure 1; Table?//Figure2 in line 159; figure 1 panels B and C in lines 160 and 161 respectively), figure S2 A and B are never cited.  

A: These are a few more very good points. Figure 1 changed to Figure 1A.    Legend to Figure 1 has been updated.    Figure S2 is now cited in section 3.2.

It would be good to briefly discuss the distribution of the selected mosses within the phylogenetic tree, for the reader to understand how representative are the results obtained. For example, how representative is the finding that the closely related Physcomitrella and Physcomitrium or Bryum and Plagiomnium exhibit similar methylation? Stating that the Pottiaceae family is an “extreme example” is not far too much considering that only two species of mosses of that family have been included in the study or they are really representative of the family?  

A: The term “extreme” was certainly ill chosen. The sentence now reads: “For this, an example …   So, overinterpretation of the statement is less likely.

References for the information regarding Chlorella vulgaris should be included (lines 280-283 of the discussion).

A: GlcNAc-transferase I, as methylation occurs on the mannose residues that becomes accessible only after action of GlcNAc-transferase I and Golgi α-mannosidase II in the canonical processing pathway.

Relevance for the possible Lewis A containing complex type glycan sequences should be discussed.

A: We can fully share and understand this reviewers curiosity. We nevertheless would rather like to refrain from unfounded speculations.

We thereby hope to have thoroughly answerd the reviewers points.